# ACIL: Analytic Class-Incremental Learning with Absolute Memorization and Privacy Protection

**Huiping Zhuang[1], Zhenyu Weng[2]\*, Hongxin Wei[3], Renchunzi Xie[3], Kar-Ann Toh[4], Zhiping Lin[2]**

[1]Shien-Ming Wu School of Intelligent Engineering, South China University of Technology, China
[2]School of Electrical and Electronic Engineering, Nanyang Technological University, Singapore
[3]School of Computer Science and Engineering, Nanyang Technological University, Singapore
[4]Department of Electrical and Electronic Engineering, Yonsei University, Korea
[1]`hpzhuang@scut.edu.cn`, [2]`{zhenyu.weng, ezplin}@ntu.edu.sg`
[3]`{hongxin001, XIER0002}@e.ntu.edu.sg`, [4]`katoh@yonsei.ac.kr`

## Abstract

Class-incremental learning (CIL) learns a classification model with training data of different classes arising progressively. Existing CIL either suffers from serious accuracy loss due to catastrophic forgetting, or invades data privacy by revisiting used exemplars. Inspired by linear learning formulations, we propose an analytic class-incremental learning (ACIL) with absolute memorization of past knowledge while avoiding breaching of data privacy (i.e., without storing historical data). The absolute memorization is demonstrated in the sense that class-incremental learning using ACIL given present data would give identical results to that from its joint-learning counterpart which consumes both present and historical samples. This equality is theoretically validated. Data privacy is ensured since no historical data are involved during the learning process. Empirical validations demonstrate ACIL's competitive accuracy performance with near-identical results for various incremental task settings (e.g., 5-50 phases). This also allows ACIL to outperform the state-of-the-art methods for large-phase scenarios (e.g., 25 and 50 phases).

## 1 Introduction

Class-incremental learning (CIL) [26, 16] trains a network phase-by-phase with training data in each phase having distinctive classes. The CIL has received an increasing popularity owing to the need to adapt learned models to unseen data classes without needing to train from scratch, allowing resource-saving and environmentally-friendly machine learning. Developing CIL is a natural call in our dynamic world where data and respective target category or task are usually available in a specific location or time slot. In addition, the CIL is intuitively motivated as it resembles real human learning processes where a person could continuously adopt knowledge of new object categories on top of the learned information.

Merits of CIL come with costs. The CIL could struggle with the notorious *catastrophic forgetting* [1], rendering the network losing grasp of the learned knowledge when accepting new tasks, which is also known as the *task-recency* bias. To mitigate the forgetting issue, several branches of CIL methods, such as the bias correction-based [2] and replay-based (or exemplar-based) [16] CIL, have been proposed. These CIL techniques are allowed to store a small number of samples from previous tasks to fight the forgetting of old knowledge. In particular, the replay-based CIL has achieved the state-of-the-art performance [12]. However, such competitive results have been obtained at the cost of revisiting the historical samples, which has brought concerns in terms of data privacy protection.

---

\*Corresponding author.

36th Conference on Neural Information Processing Systems (NeurIPS 2022).

**Data Privacy in CIL.** Data privacy is becoming more of value in our interconnected modern world, which naturally applies to CIL problems asking for exemplar-free learning. Note that the "privacy" in CIL (i.e., cannot re-use past exemplars) may be different from the definition of other fields (such as data encryption). The increasing concern of data privacy contradicts many existing CIL techniques, such as the replay-based CIL. Several methods from the regularization-based CIL [9] respect privacy as they only impose regularization terms on the loss functions. However, without re-accessing the trained samples, their accuracy performance cannot compete with that of the replay-based CIL. Another exemplar-free CIL branch is the generative adversarial network (GAN)-based learning [21], which preserves privacy by generating historical samples using GANs. This CIL relies heavily on GANs' performance and has not been tested in challenging datasets such as ImageNet [3].

In summary, existing CIL techniques either invade data privacy (e.g., replay-based CIL) or cannot provide satisfactory accuracy performance (e.g., regularization-based CIL). In addition, as the forgetting issue persists (though mitigated), CIL's performance experiences a significant degradation as learning phases increase, a pattern shared by many existing CIL techniques (e.g., [4]). The performance degradation escalates in large-phase scenarios—a learning scenario with a large number of learning phases for increment (e.g., 50 learning phases [24]). This has motivated us to find new CIL methods that well tackles the catastrophic forgetting without invading data privacy.

In this paper, we propose an analytic class-incremental learning (ACIL) to handle the issue of forgetting and privacy invasion in CIL. The ACIL is inspired by *analytic learning* [29, 5], a technique that formulates network training into learning of linear stacks. The analytic learning component allows the ACIL to conduct CIL in a recursive learning manner that can absolutely memorize the knowledge of every historical sample (i.e., address catastrophic forgetting) while avoiding the breach of data privacy (i.e., without storing any past data). The key contributions are summarized as follows.

• We introduce the ACIL, which holds absolute memorization of previous knowledge when accepting new tasks.

• The ACIL does not store past samples, thereby achieving data privacy protection, a rare but valuable CIL property.

• We provide theoretical validation of ACIL's absolute memorization, showing that the CIL using ACIL given present data provides an identical result to that from its joint-learning counterpart that adopts data from both present and historical phases.

• Experiments on benchmark datasets show that the ACIL gives competitive CIL results that do not degrade over increment of data classes during learning phases. In particular, it outperforms the state-of-the-arts by a considerable margin for relatively large-phase scenarios (e.g., 25 or 50 phases).

## 2 Related Works

### 2.1 Class-Incremental Learning

**Bias correction-based** CIL mainly tries to address the task-recency bias. The end-to-end incremental learning [2] reduces the bias by introducing a balance training stage where only an equal number of samples for each class is used. The bias correction (BiC) [23] includes an additional trainable layer which aims to correct the bias. The method named LUCIR proposed in [6] fights the bias by changing the softmax layer into a cosine normalization one.

**Replay-based** CIL stores a small subset of data from the previously accessed tasks to reinforce the network's memory of old knowledge. This CIL branch quickly draws attention due to the appealing ability to resist the catastrophic forgetting. For instance, the PODNet [4] adopts an efficient spatial-based distillation loss to reduce forgetting, with a focus on the large-phase setting, achieving reasonably good results. The AANets [11] employs a new architecture containing a stable block and a plastic block to balance the stability and plasticity. On top of the replay-based CIL, methods exploring exemplar storing techniques [13] are also fruitful. For instance, the reinforced memory management (RMM) [12] seeks a dynamic memory management using reinforcement learning. By plugging it onto PODNet and AANets, the RMM attains a state-of-the-art performance.

**Regularization-based** CIL imposes additional constraints on the loss functions to avoid forgetting. The regularization can be imposed on weights by estimating the parameters' importance so relevant weights do not drift significantly. The elastic weight consolidation (EWC) [8] captures the prior

importance using an diagonally approximated Fisher information matrix. The EWC is improved by [10] through finding a better approximation of the Fisher information matrix. The regularization can also be imposed on activations to prevent activation drift, which outperforms its weight-regularization counterpart in general. The learning without forgetting (LwF) [9] prevents activations of the old network from drifting while learning new tasks. The less-forgetting learning [7] penalizes the activation difference except the fully-connected layer.

**GAN-based** CIL replays past samples by generating them using GANs. The deep generative replay [17] generates synthetic samples using an unconditioned GAN. It is later improved by memory replay GAN [22] adopting a label-conditional GAN. In general, the GAN-based CIL relies heavily on GAN's generative performance, and is only tested on relatively small datasets, such as MNIST.

The bias correction-based and replay-based CILs allow storing exemplars, leading to privacy invasion. The exemplar-free methods (e.g., regularization-based and GAN-based CIL) do not give competitive results. Our ACIL can memorize historical knowledge without re-accessing the data from previous tasks, allowing it to perform CIL with absolute memorization and privacy reservation.

## 2.2 Analytic Learning

The analytic learning has been developed to circumvent limitations imposed by back-propagation (BP), such as gradient vanishing/exploding, divergence during iteration and long training time (i.e., need many epochs). The analytic learning also goes by other names such as *pseudoinverse learning* [5] due to the use of matrix inverse. The analytic learning starts with the shallow learning. One quick example is the radial basis network [15], which trains the parameters using a least-squares (LS) estimation after conducting a kernel transformation in the first layer. The multilayer analytic learning [18, 20] converts the nonlinear network learning into linear segments that can be solved adopting LS techniques in a one-epoch training style. For instance, the dense pseudoinverse autoencoder [19] trains a stacked autoencoder layer-by-layer by concatenating shallow and deep features using LS solutions. The analytic learning could experience out-of-memory issue as the weights are learned involving the entire dataset at once. Such a memory issue can be addressed by the block-wise recursive Moore-Penrose learning (BRMP) [28] by replacing the joint learning with a recursive one. This much resembles the replacement of gradient descent with stochastic gradient descent to reduce memory usage, but differs in that the BRMP can exactly reproduce its joint-learning result.

The analytic learning and its recursive formulation (e.g., BRMP) brings inspiration to the CIL realm. The BRMP can stream new samples to update the weight without weakening the impact of previous samples. This matches the ACIL's need for the enhanced memorization of previously trained data. By bridging the analytic learning and its recursive formulation, our ACIL can be built to absolutely remember historical samples, without needing to re-access them.

## 3 The Proposed Method

This section presents the algorithmic details of ACIL, including a *base training agenda* and a *CIL agenda*. Our presentation of ACIL is mainly rooted in CNNs which contain a CNN backbone (feature extractor) followed by a fully-connected network (FCN) layer (classifier) for classification problems. An overview of ACIL is depicted in Figure 1.

### 3.1 The Base Training Agenda

The base training agenda of ACIL has two stages in a sequential oder, namely a base training via BP and an analytic re-alignment base training (ARaBT), which are illustrated in Figure 1(a) and Figure 1(b) respectively.

**Base Training via BP**. The first stage (see Figure 1(a)) of the base training agenda duplicates the conventional BP training on the base dataset. That is, the network is trained with a BP-based iteration algorithm (e.g., SGD with momentum) for multiple epochs with an appropriate learning rate scheduler (e.g., step decay scheduler). Let $W_{\text{CNN}}$ and $W_{\text{FCN}}$ represent the weights for the CNN backbone and the FCN classifier. After the BP-base base training, given an input $X$, the output of the network is

$$Y = f_{\text{softmax}}(f_{\text{flat}}(f_{\text{CNN}}(X, W_{\text{CNN}}))W_{\text{FCN}})$$

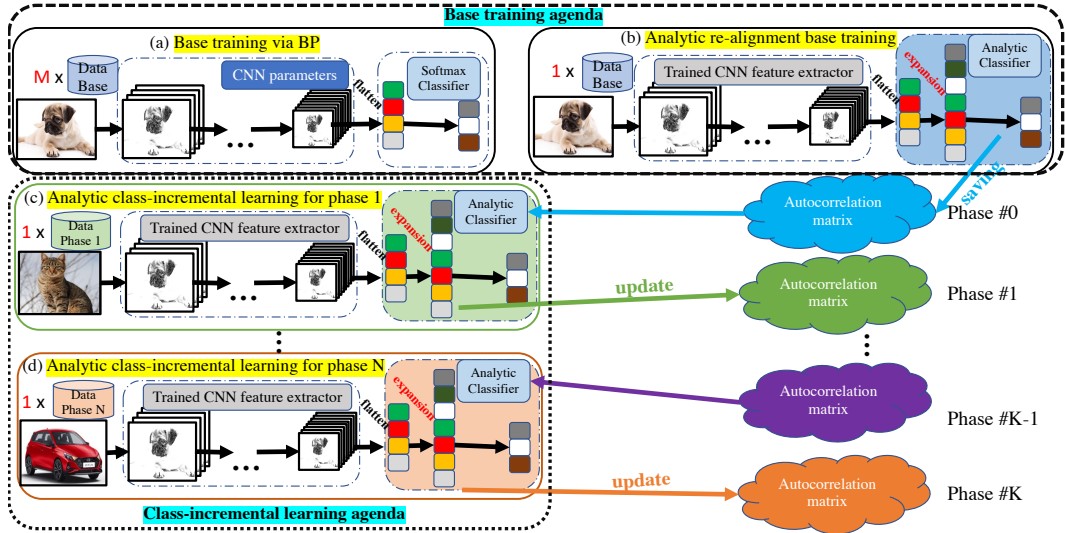

**Figure 1:** The ACIL begins with the **base training agenda**: (a) training a network with BP-based iteration method for $M$ epochs on the base dataset, followed by (b) ARaBT for 1 epoch only on the same dataset, which expands the FCN dimension to enhance feature extraction. Subsequently, (c-d) the **CIL agenda** is conducted in a recursive manner adopting the dataset (train for 1 epoch) at the current phase only and a correlation matrix (see definition in (8)) encrypted with historical information.

where $f_{\text{CNN}}(\boldsymbol{X}, \boldsymbol{W}_{\text{CNN}})$ indicates the CNN backbone output with an $\boldsymbol{X}$ passing through it; $f_{\text{flat}}$ is a flattening operator, reshaping a training sample into a 1-D vector; $f_{\text{softmax}}$ is the softmax function.

**Analytic Re-alignment Base Training**. The second stage of base training (see Figure 1(b)), the ARaBT, is the key to the formulation of ACIL. In this stage, the ARaBT "re-aligns" the network's learning to match the learning dynamics of an analytic learning.

Prior to our development, some definitions related to CIL are introduced. A $K$-phase CIL indicates that a network is trained for $K$ phases where training data of each phase comes with different classes. Let $\mathcal{D}_k^{\text{train}} \sim \{\boldsymbol{X}_k^{\text{train}}, \boldsymbol{Y}_k^{\text{train}}\}$ and $\mathcal{D}_k^{\text{test}} \sim \{\boldsymbol{X}_k^{\text{test}}, \boldsymbol{Y}_k^{\text{test}}\}$ be the training and testing datasets at phase $k$ ($k = 1, 2, \ldots, K$). $\boldsymbol{X}_k \in \mathbb{R}^{N_k \times w \times h \times c}$ (e.g., images with a shape of $w \times h \times c$) and $\boldsymbol{Y}_k \in \mathbb{R}^{N_k \times d_{y_k}}$ (with phase $k$ including $d_{y_k}$ classes) are stacked input and label (one-hot) tensors. Here $\mathcal{D}_0^{\text{train}} \sim \{\boldsymbol{X}_0^{\text{train}}, \boldsymbol{Y}_0^{\text{train}}\}$ represents the base training set utilized to conduct the ARaBT.

The first step is to extract the feature matrix (denoted by $\boldsymbol{X}_0^{(\text{cnn})}$) by feeding the input tensor $\boldsymbol{X}_0^{\text{train}}$ through the trained CNN backbone, followed by a flattening operation, i.e.,

$$\boldsymbol{X}_0^{(\text{cnn})} = f_{\text{flat}}(f_{\text{CNN}}(\boldsymbol{X}_0^{\text{train}}, \boldsymbol{W}_{\text{CNN}})) \tag{1}$$

where $\boldsymbol{X}_0^{(\text{cnn})} \in \mathbb{R}^{N_0 \times d_{\text{cnn}}}$. Instead of building one FCN layer to map the feature onto the classification terminal, we conduct a *feature expansion* (FE) process by inserting an additional FCN layer which expands the feature space into a higher one. That is, the feature $\boldsymbol{X}_0^{(\text{cnn})}$ is expanded to $\boldsymbol{X}_0^{(\text{fe})}$ as follows

$$\boldsymbol{X}_0^{(\text{fe})} = f_{\text{act}}(f_{\text{flat}}(f_{\text{CNN}}(\boldsymbol{X}_0^{\text{train}}, \boldsymbol{W}_{\text{CNN}}))\boldsymbol{W}_{\text{fe}}) = f_{\text{act}}(\boldsymbol{X}_0^{(\text{cnn})}\boldsymbol{W}_{\text{fe}}) \tag{2}$$

where $\boldsymbol{X}_0^{(\text{fe})} \in \mathbb{R}^{N_0 \times d_{(\text{fe})}}$ with $d_{(\text{fe})}$ being the *expansion size* (with $d_{\text{cnn}} \leq d_{(\text{fe})}$). $f_{\text{act}}$ is an activation function (we adopt ReLU in this paper), and $\boldsymbol{W}_{\text{fe}}$ is the FE matrix expanding the CNN-extracted feature. The need for FE process can be justified by the fact that analytic-learning methods require more parameters to achieve their maximum performance. For the FE matrix, we determine $d_{(\text{fe})}$ with a very simple trick by drawing every element from a normal distribution. Such a randomization technique has been shown to capture useful information for classification problems (e.g., see [5, 29]).

Finally, the expanded feature $\boldsymbol{X}_0^{(\text{fe})}$ is mapped onto the label matrix $\boldsymbol{Y}_0^{\text{train}}$ using a linear regression procedure via solving

$$\underset{\boldsymbol{W}_{\text{FCN}}^{(0)}}{\arg\min} \quad \left\|\boldsymbol{Y}_0^{\text{train}} - \boldsymbol{X}_0^{(\text{fe})}\boldsymbol{W}_{\text{FCN}}^{(0)}\right\|_F^2 + \gamma \left\|\boldsymbol{W}_{\text{FCN}}^{(0)}\right\|_F^2 \tag{3}$$

where $\|\cdot\|_F$ indicates the Frobenius norm, and $\gamma$ regularizes the above objective function. Also, $\cdot^{\mathrm{T}}$ is the matrix transpose operator. The optimal solution to (3) can be found in

$$\hat{\boldsymbol{W}}_{\mathrm{FCN}}^{(0)} = (\boldsymbol{X}_0^{(\mathrm{fe})\mathrm{T}}\boldsymbol{X}_0^{(\mathrm{fe})} + \gamma\boldsymbol{I})^{-1}\boldsymbol{X}_0^{(\mathrm{fe})\mathrm{T}}\boldsymbol{Y}_0^{\mathrm{train}} \tag{4}$$

where $\hat{\boldsymbol{W}}_{\mathrm{FCN}}^{(0)}$ indicates the estimated FCN weight of the final classifier layer.

## 3.2 The Class-Incremental Learning Agenda

With the network learning aligned with the analytic learning (see (4)), we may proceed to CIL in an analytic learning fashion. To this end, assume that we are given $\mathcal{D}_0^{\mathrm{train}}, \dots, \mathcal{D}_{k-1}^{\mathrm{train}}$, the learning problem in (3) can be extended to

$$\underset{\boldsymbol{W}_{\mathrm{FCN}}^{(k-1)}}{\operatorname{argmin}} \left\| \begin{bmatrix} \boldsymbol{Y}_0^{\mathrm{train}} & \boldsymbol{0} & \boldsymbol{0} \dots & \boldsymbol{0} \\ \boldsymbol{0} & \boldsymbol{Y}_1^{\mathrm{train}} & \boldsymbol{0} \dots & \boldsymbol{0} \\ & & \vdots & \\ \boldsymbol{0} & \boldsymbol{0} & \dots \boldsymbol{Y}_{k-1}^{\mathrm{train}} \end{bmatrix} - \begin{bmatrix} \boldsymbol{X}_0^{(\mathrm{fe})} \\ \boldsymbol{X}_1^{(\mathrm{fe})} \\ \vdots \\ \boldsymbol{X}_{k-1}^{(\mathrm{fe})} \end{bmatrix} \boldsymbol{W}_{\mathrm{FCN}}^{(k-1)} \right\|_F^2 + \gamma \left\| \boldsymbol{W}_{\mathrm{FCN}}^{(k-1)} \right\|_F^2 \tag{5}$$

where

$$\boldsymbol{X}_i^{(\mathrm{fe})} = f_{\mathrm{act}}(f_{\mathrm{flat}}(f_{\mathrm{CNN}}(\boldsymbol{X}_i^{\mathrm{train}}, \boldsymbol{W}_{\mathrm{CNN}}))\boldsymbol{W}_{\mathrm{fe}}). \tag{6}$$

Note that the stacked label matrix in (5) has a sparse structure due to the fact that datasets from different phases are mutually exclusive. The solution to (5) can be written as

$$\hat{\boldsymbol{W}}_{\mathrm{FCN}}^{(k-1)} = \left( \sum_{i=0}^{k-1} \boldsymbol{X}_i^{(\mathrm{fe})\mathrm{T}}\boldsymbol{X}_i^{(\mathrm{fe})} + \gamma\boldsymbol{I} \right)^{-1} \begin{bmatrix} \boldsymbol{X}_0^{(\mathrm{fe})\mathrm{T}}\boldsymbol{Y}_0 & \dots & \boldsymbol{X}_{k-1}^{(\mathrm{fe})\mathrm{T}}\boldsymbol{Y}_{k-1} \end{bmatrix} \tag{7}$$

where $\hat{\boldsymbol{W}}_{\mathrm{FCN}}^{(k-1)} \in \mathbb{R}^{d_{(\mathrm{fe})} \times \sum_{i=1}^{k-1} d_{y_i}}$ with a column size proportional to $k$.

Equation (7) gives an LS-based analytical solution for joint learning on $\mathcal{D}_{0:k-1}^{\mathrm{train}}$. The goal of ACIL is to calculate the analytical solution that satisfies (5) at phase $k$ based on $\hat{\boldsymbol{W}}_{\mathrm{FCN}}^{(k-1)}$ given $\mathcal{D}_k^{\mathrm{train}}$ without any samples from $\mathcal{D}_{0:k-1}^{\mathrm{train}}$. Specifically, we aim to obtain $\hat{\boldsymbol{W}}_{\mathrm{FCN}}^{(k)}$ recursively based on $\hat{\boldsymbol{W}}_{\mathrm{FCN}}^{(k-1)}$ and data $\boldsymbol{X}_k^{(\mathrm{fe})}, \boldsymbol{Y}_k^{\mathrm{train}}$ that are available only at the current learning phase. However, the updated weight $\hat{\boldsymbol{W}}_{\mathrm{FCN}}^{(k)}$ must satisfy the joint learning in (5) given $\mathcal{D}_{0:k}^{\mathrm{train}}$. Let

$$\boldsymbol{R}_{k-1} = \left( \sum_{i=0}^{k-1} \boldsymbol{X}_i^{(\mathrm{fe})\mathrm{T}}\boldsymbol{X}_i^{(\mathrm{fe})} + \gamma\boldsymbol{I} \right)^{-1} \tag{8}$$

be the *regularized feature autocorrelation matrix* (RFAuM) at learning phase $k-1$. Then our solution can be summarized in the following Theorem.

**Theorem 3.1.** The FCN weight recursively obtained by

$$\hat{\boldsymbol{W}}_{\mathrm{FCN}}^{(k)} = \begin{bmatrix} \hat{\boldsymbol{W}}_{\mathrm{FCN}}^{(k-1)} - \boldsymbol{R}_k\boldsymbol{X}_k^{(\mathrm{fe})\mathrm{T}}\boldsymbol{X}_k^{(\mathrm{fe})}\hat{\boldsymbol{W}}_{\mathrm{FCN}}^{(k-1)} & \boldsymbol{R}_k\boldsymbol{X}_k^{(\mathrm{fe})\mathrm{T}}\boldsymbol{Y}_k^{\mathrm{train}} \end{bmatrix} \tag{9}$$

is identical to that obtained by (7) at phase $k$. The RFAuM $\boldsymbol{R}_k$ can also be recursively calculated by

$$\boldsymbol{R}_k = \boldsymbol{R}_{k-1} - \boldsymbol{R}_{k-1}\boldsymbol{X}_k^{(\mathrm{fe})\mathrm{T}}(\boldsymbol{I} + \boldsymbol{X}_k^{(\mathrm{fe})}\boldsymbol{R}_{k-1}\boldsymbol{X}_k^{(\mathrm{fe})\mathrm{T}})^{-1}\boldsymbol{X}_k^{(\mathrm{fe})}\boldsymbol{R}_{k-1} \tag{10}$$

*Proof.* See the supplementary materials. $\square$

As shown in Theorem 3.1, the proposed ACIL constructs a recursive update of the FCN weight matrix without any loss of historical information. One can first conduct the base training agenda on the base dataset (e.g., compute $\hat{\boldsymbol{W}}_{\mathrm{FCN}}^{(0)}$), and perform CIL afterwards adopting the recursive formulation to obtain $\hat{\boldsymbol{W}}_{\mathrm{FCN}}^{(k)}$ for $k > 0$. The computational steps of ACIL is summarized in Algorithm 1.

**Absolute Memorization**. As observed in Theorem 3.1, the CIL in (9) yields an identical result to that of the joint learning in (7). This allows the ACIL to operate with *absolute memorization* in the sense that the recursive formulation (i.e., the incremental learning) gives the same answer as the one

---

**Algorithm 1** ACIL

---

**Base training agenda:** with $\mathcal{D}_0^{\text{train}}$.
1. Conventional training with BP on base dataset.
2. ARaBT: i) Obtain feature matrix with (2); ii) Obtain re-aligned weight $\hat{\boldsymbol{W}}_{\text{FCN}}^{(0)}$ with (4). iii) Save RFAuM $\boldsymbol{R}_0$.

**CIL agenda:**
**for** $k = 1$ **to** $K$ (with $\mathcal{D}_k^{\text{train}}$, $\hat{\boldsymbol{W}}_{\text{FCN}}^{(k-1)}$ and $\boldsymbol{R}_{k-1}$) **do**
    i) Obtain feature matrix with (6);
    ii) Update RFAuM $\boldsymbol{R}_k$ with (10);
    iii) Update weight matrix $\hat{\boldsymbol{W}}_{\text{FCN}}^{(k)}$ with (9);
**end for**

---

obtained by its joint analytic learning counterpart. Such an absolute memorization differentiates our method from the existing CIL techniques that are struggling to fight the forgetting issue. To the best of our knowledge, the ACIL is the first CIL that achieves absolute memorization.

**Data Privacy Protection**. Another benefit of our ACIL lies in privacy protection. Algorithm 1 shows that, during the CIL agenda, no historical samples are granted. Instead, the $\boldsymbol{R}_k$ is cached to encrypt information for historical samples. However, it is impossible to reverse-engineer the process to obtain the original samples based on the $\boldsymbol{R}_k$ only, avoiding possible breaching of data privacy. This is an attractive feature as data privacy has attracted increasing concern in the CIL community.

Although methods in regularization-based CIL (e.g., LwF) could also protect data privacy, the accuracy performance is less ideal. In comparison, as latter shown in the experiments (e.g., Table 1), the proposed ACIL preserves data privacy while achieving very competitive results. In addition, the RFAuM holds a fixed shape (i.e., a square matrix of $\mathbb{R}^{d_{\text{ce}} \times d_{\text{ce}}}$) regardless of the sample size. This takes up less storage room than that of the replay-based CIL.

**An Analytic-Learning Branch of CIL.** We may categorize the ACIL into a new branch (i.e., analytic-learning branch) of CIL. Unlike other branches, the ACIL does not forget any historical information at all. It also attains privacy protection, a rare but valuable CIL feature. Even with several appealing features, the ACIL is naturally not as powerful as the BP-based joint learning. The ACIL is facilitated but also constrained by the fact that it freezes the training of CNN weights. As seen in (6), the feature matrix $\boldsymbol{X}_i^{(\text{fe})}$ during the CIL agenda is constructed purely based on a transfer learning w.r.t. the CNN backbone trained on the base dataset. That is, the ACIL extracts the feature of new task classes using a somewhat obsolete feature extractor. This would lead to certain performance drop. However, we would argue that the benefits of ACIL greatly outweigh the potential accuracy loss. Our argument are well supported by the experiments, displaying very competitive results using ACIL.

## 4 Experiments

We evaluate the proposed ACIL on CIFAR-100, ImageNet-Subset and ImageNet-Full datasets which are benchmark datasets for CIL. We compare the ACIL with several state-of-the-art CIL techniques, including LwF [9], EWC [8], Semantic Drift Compensation (SDC) [25], BIC [23], iCaRL [16], LUCIR [6], Mnemonics [13], PODNet [4], AANets [11] and RMM [12]. GAN-based CIL is not included as it is only tested on less challenging datasets (e.g., MNIST) and its performance relies heavily on the GAN training.

The LwF adopts distillation-based loss functions to prevent forgetting. The EWC uses Fisher information matrix. The SDC studies and compensates the semantic drift of features. The LwF, EWC, SDC and the proposed ACIL belong to the privacy-preserving methods. The other methods, i.e., BIC, iCaRL, Mnemonics, PODNet, AANets and RMM, are replay-based methods, requiring storage of past exemplars.

### 4.1 Datasets and Implementation Details

**Datasets.** CIFAR-100 contains 100 classes of $32 \times 32$ color images with each class having 500 and 100 images for training and testing respectively. ImageNet-Full has 1000 classes, and 1.3 million

**Table 1:** Comparison of $\bar{\mathcal{A}}$ and $\mathcal{F}$ among compared methods. The ACIL adopts $d_{y_k} = $ 8k, 15k, 15k ("1k"=1000) on CIFAR-100, ImageNet-Subset and ImageNet-Full respectively. The ACIL, LwF, EWC and SDC do not keep old data while other compared methods adopt the same replay settings (e.g., [4, 16]) by reserving 20 exemplars per old class. Results for $\bar{\mathcal{A}}(\%)$ are duplicated from [11]) except for the 3-combo method "POD+AANets+RMM" which is copied from the RMM paper [12] (its ImageNet-Full results are not listed due to no ImageNet option in the source code). Results for $\mathcal{F}(\%)$ are cloned from [13]. The strict-memory setting results can be found in Table A in the supplementary materials.

| Metric | Method | Privacy | CIFAR-100 | | | | ImageNet-Subset | | | | ImageNet-Full | | | |
|---|---|---|---|---|---|---|---|---|---|---|---|---|---|---|
| | | | K=5 | 10 | 25 | 50 | K=5 | 10 | 25 | 50 | K=5 | 10 | 25 | 50 |
| $\bar{\mathcal{A}}(\%)$ | LwF (TPAMI 2018) | ✓ | 49.59 | 46.98 | 45.51 | - | 53.62 | 47.64 | 44.32 | - | 51.50 | 46.89 | 43.14 | - |
| | EWC (PNAS) | ✓ | 34.01 | 32.33 | - | - | 42.35 | 26.76 | - | - | - | - | - | - |
| | SDC (CVPR 2020) | ✓ | 55.96 | 56.56 | - | - | - | 62.97 | - | - | - | - | - | - |
| | BiC (CVPR 2019) | ✗ | 59.36 | 54.20 | 50.00 | - | 70.07 | 64.96 | 57.73 | - | 62.65 | 58.72 | 53.47 | - |
| | iCaRL (CVPR 2017) | ✗ | 57.12 | 52.66 | 48.22 | - | 65.44 | 59.88 | 52.97 | - | 51.50 | 46.89 | 43.14 | - |
| | LUCIR (CVPR 2019) | ✗ | 63.17 | 60.14 | 57.54 | - | 70.84 | 68.32 | 61.44 | - | 64.45 | 61.57 | 56.56 | - |
| | PODNet (ECCV 2020) | ✗ | 64.83 | 63.19 | 60.72 | 57.98 | 75.54 | 74.33 | 68.31 | 62.48 | 66.95 | 64.13 | 59.17 | - |
| | LUCIR+Mnemonics (CVPR 2020) | ✗ | 64.95 | 63.25 | 63.70 | - | 73.30 | 72.17 | 71.50 | - | 66.15 | 63.12 | 63.08 | - |
| | POD+AANets (CVPR 2021) | ✗ | 66.31 | 64.31 | 62.31 | - | 76.96 | 75.58 | 71.78 | - | **67.73** | **64.85** | 61.78 | - |
| | POD+AANets+RMM (NeurIPS 2021) | ✗ | **68.36** | **66.67** | 64.12 | - | **79.50** | **78.11** | **75.01** | - | - | - | - | - |
| | ACIL | ✓ | 66.30 | 66.07 | **65.95** | **66.01** | 74.81 | 74.76 | 74.59 | **74.13** | 65.34 | 64.84 | **64.63** | **64.35** |
| $\mathcal{F}(\%)$ | LwF (TPAMI 2018) | ✓ | 43.36 | 43.58 | 41.66 | - | 55.32 | 57.00 | 55.12 | - | 48.70 | 47.94 | 49.84 | - |
| | iCaRL (CVPR 2017) | ✗ | 57.12 | 34.10 | 36.48 | - | 43.40 | 45.84 | 47.60 | - | 26.03 | 33.76 | 38.80 | - |
| | BiC (CVPR2019) | ✗ | 31.42 | 32.50 | 34.60 | - | 27.04 | 31.04 | 37.88 | - | 25.06 | 28.34 | 33.17 | - |
| | LUCIR (CVPR 2019) | ✗ | 18.70 | 21.34 | 26.46 | - | 31.88 | 33.48 | 35.40 | - | 24.08 | 27.29 | 30.30 | - |
| | LUCIR+Mnemonics (CVPR 2020) | ✗ | 11.64 | 10.90 | 9.96 | - | 10.20 | 9.88 | 11.76 | - | 13.63 | 13.45 | 14.40 | - |
| | ACIL | ✓ | **9.00** | **9.72** | **9.28** | **9.32** | **3.91** | **3.40** | **3.20** | **3.43** | **2.75** | **3.45** | **3.31** | **3.40** |

images for training with 50,000 images for testing. ImageNet-Subset, in particular, is constructed by selecting 100 specific classes from ImageNet-Full based on what defined in [4].

**Network Architecture.** The architectures for CIL in the experiments are ResNet-32 on CIFAR-100 and ResNet-18 on both ImageNet-Full and its subset. These two architectures are commonly adopted for CIL performance comparison. Our ACIL imposes a slight change (i.e., inserts an expansion FCN layer), but the CNN backbones are identical to those from the selected ResNet architectures.

**Training Details.** For conventional BP training in the base training agenda, we train the network using SGD for 160 (90) epochs for ResNet-32 (ResNet-18). The learning rate starts at 0.1 and it is divided by 10 at epoch 80 (30) and 120 (60). We adopt a momentum of 0.9 and weight decay of $5 \times 10^{-4}$ ($1 \times 10^{-4}$) with a batch size of 128. The input data are augmented with random cropping, random horizontal flip and normalizing. For fair comparison, this base training setting is identical to that of many CIL methods (e.g., [6, 13]). For the ARaBT and ACIL's incremental learning steps, no data augmentation is adopted, and the training ends within only one epoch. The results for the ACIL are measured by the average of 3 runs on an RTX 2080Ti GPU workstation. Note that in ACIL, the CNN is only trained during the BP base training phase. After that, the parameters of the CNN backbone are fixed during the incremental phases (i.e, phase #1 to #K) including the ARaBT.

**CIL Protocol.** We follow the protocol adopted in [4, 11]. The network is first trained (i.e., phase #0) on the base dataset containing half of the full classes from the original dataset. Subsequently, the network gradually learns the remaining classes evenly for $K$ phases (i.e., $K$-phase CIL), with the dataset in each phase containing disjoint classes from one another. Most existing methods only report results for $K = 5, 10, 25$. We include $K = 50$ as well to validate ACIL's absolute memorization.

## 4.2 Evaluation Metric

Two metrics are adopted to evaluate the ACIL. The overall accuracy performance is evaluated by the *average incremental accuracy* (or average accuracy) $\bar{\mathcal{A}}$ (%): $\bar{\mathcal{A}} = \frac{1}{K+1}\sum_{k=0}^{K}\mathcal{A}_k$ where $\mathcal{A}_k$ indicates the average test accuracy of the network incrementally trained at phase $k$ by testing it on $\mathcal{D}_{0:k}^{\text{test}}$. The $\bar{\mathcal{A}}$ evaluates the overall performance of CIL algorithms. A higher $\bar{\mathcal{A}}$ score is preferred when evaluating CIL algorithms. The other evaluation metric is the *forgetting rate* $\mathcal{F}$ (%) defined in [13]: $\mathcal{F} = A_K^Z - A_0^Z$ where $A_k^Z$ denotes the average accuracy at phase $k$ by testing it on $\mathcal{D}_0^{\text{test}}$. The forgetting rate reveals the degree to which a CIL method forgets the base classes. Hence, it is a good indicator to evaluate CIL's forgetting issue.

## 4.3 Result Comparison

We tabulate the average incremental accuracy $\bar{\mathcal{A}}$ and the forgetting rate $\mathcal{F}$ from the compared methods in Table 1. As shown in the upper panel, overall, the "combo" CIL techniques—techniques

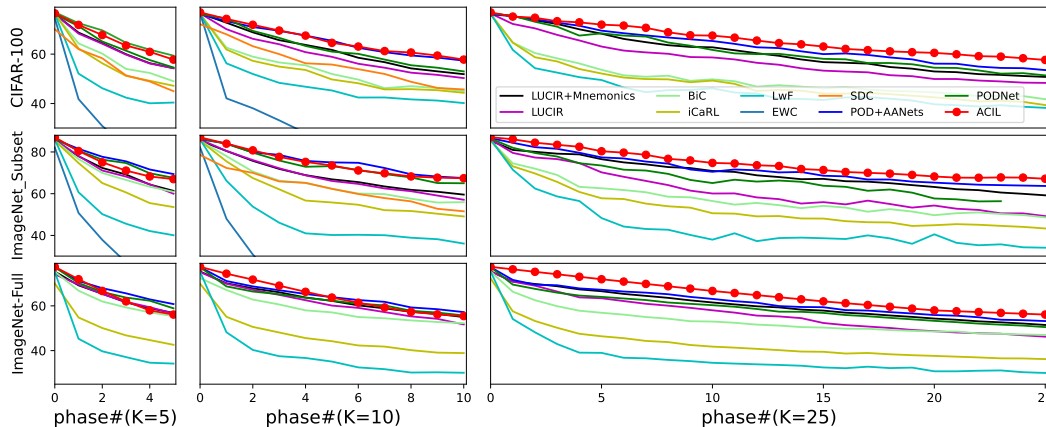

**Figure 2:** Avg. accuracy w.r.t. phase. The RMM curve is not included as its source code is only applicable for strict-memory settings.

that combine more than one CIL methods—give very competitive $\bar{\mathcal{A}}$ scores. In particular, the "POD+AANets+RMM" combo, which incorporates PODNet [4], AANets [11] into RMM [12], obtains the most competitive results that can be treated as current the state-of-the-art counterpart.

As shown in the upper panel of Table 1, the ACIL gives a slightly worse average accuracy for 5-phase CIL in general. However, ACIL's performance catches up with those of the state-of-th-arts as $K$ increases, and begins to lead for $K \geq 25$ (i.e., large-phase CIL scenarios). For instance, for 5-phase CIL, the ACIL gives an accuracy of 66.30% on CIFAR-100, which is slightly worse than the results from several combo techniques such as the "POD+AANets" combo (66.31%) by 0.01% and the "POD+AANets+RMM" combo (68.36%) by 2.06%. However, for 25-phase CIL, the ACIL (with 65.95%) outperforms these combo methods, e.g., outperforming the second best by 1.83% (64.12% from the "POD+AANets+RMM" combo). Such an overtaking pattern is naturally expected owing to ACIL's absolute memorization. That is, the accuracy of ACIL remains unchanged for different $K$ values, while other CIL methods experience various levels of forgetting issue that intensifies as $K$ increases. Note that there could be a very mild $\bar{\mathcal{A}}$ degradation from ACIL as $K$ increase (e.g., 66.30%→65.95). Although theoretically the ACIL should give identical results regardless of $K$, the possible mild drop is likely caused by quantization errors since large $K$ indicates more computation rounds hence more quantization operations (e.g., see TABLE VI [28]).

This pattern on CIFAR-100 is quite consistent with those on ImageNet-Subset and ImageNet-Full. On ImageNet-Full, the ACIL begins to outperform the compared methods for $K \geq 10$, and leads (with 64.63%) the second best result (63.08% from the "LUCIR+Mnemonics" combo) by 1.55% for 25-phase learning. On ImageNet-Subset, the ACIL falls behind the "POD+AANets+RMM" comb even for 25-phase learning, but the gap is very small (74.59% v.s. 75.01%). The overtaking pattern is further detailed in Figure 2.

In addition, we report the 50-phase CIL for the proposed ACIL. As expected, the average accuracies (i.e., 66.01%, 74.13% and 64.35% on CIFAR-100, ImageNet-Subset and ImageNet-Full) are very close to those trained with $K = 5, 10$ or $25$. This allows the ACIL to further outperform the compared methods that are not specializing in large-phase incremental problems. Although the PODNet also aims at large-phase problems, its performance cannot compete with ACIL's (57.98% v.s. 66.01% on CIFAR-100, and 62.48% v.s. 74.13% on ImageNet-Subset).

**Why ACIL Performs Well.** Conventionally, the analytic learning cannot compete with BP [28]. However, if the feature extractor (e.g., CNN layers) is pre-trained with BP with the classifier head designed by analytic learning related techniques, the performance can catch up [14]. Such a scenario fits well in the CIL procedure in this paper, explaining why our ACIL performs well.

The expansion size $d_{\text{(fe)}}$ from the FE process has a huge impact on the CIL performance. As plotted in Figure 3, the $\bar{\mathcal{A}}$ on ImageNet-100 and ImageNet-Full increases with larger $d_{\text{(fe))}}$. The performance on CIFAR-100 starts to decline for $d_{\text{(fe))}} > 10k$, likely because the expansion ratio for ResNet-32 case is unreasonably large (e.g., $d_{\text{(fe))}}/d_{\text{(cnn)}} = 15k/64$) compared with that of ResNet-18 (15k/512) on ImageNet datasets. As observed in Figure 3, the ACIL on ImageNet datasets should work better with

even larger $d_{\text{(fe)}}$, but our 11GB GPU experiences memory leak. Still, the expansion up to $d_{\text{(fe)}} = 15\text{k}$ allows the ACIL to give very competitive results (see Table 1).

The forgetting rate $\mathcal{F}$ is also presented in the bottom panel of Table 1. Our ACIL demonstrates the lowest $\mathcal{F}$ scores on all the three benchmark datasets. This is a further evidence supporting our absolute-memorization claim. Note that the absolute memorization does not lead to $\mathcal{F} = 0$. Even a healthy joint learning would reduce the performance on the base classes. This can be explained by the example as follows. Let $\mathcal{M}_{50}$ and $\mathcal{M}_{100}$ be the two networks jointly trained on the base 50 classes and the 100 full classes from CIFAR-100 respectively. Testing $\mathcal{M}_{100}$ on the base dataset would still experience performance loss compared with that obtained by testing $\mathcal{M}_{50}$ on the base dataset, i.e., $\mathcal{F} > 0$. Hence, although our ACIL perfectly remembers the pass samples, non-zero forgetting rate still applies when incrementally learning new classes. Nonetheless, the forgetting rate has been shown to be much lower than the existing CIL methods. For instance, for 5-phase learning on ImageNet-Full, the "LUCIR+Mnemonics" combo exhibits 13.63% forgetting, but our ACIL only has 2.75%! Yet, the low $\mathcal{F}$ score does not suggest strong resistance for learning new tasks since the average accuracy has been shown to be comparable to state-of-the-art results (see upper panel of Table 1). That is, the ACIL bears a relatively good stability-plasticity balance.

**Data Privacy Protection.** Apart from the competitive incremental accuracy, the ACIL is in strong support of data privacy. As indicated in Algorithm 1, the incremental learning surrenders any samples from previous tasks, allowing data privacy across learning phases or platforms. As shown in Table 1, privacy-preserving CIL (e.g., LwF, EWC and SDC) suffers much more intensively (e.g., 45.51% from LwF of 25-phase CIL on CIFAR-100) than the replay-based CIL (e.g., 64.12% from RMM combo of 25-phase CIL on CIFAR-100). Our ACIL maintains the privacy while providing comparable or better accuracy performance (e.g., 65.95% of 25-phase CIL on CIFAR-100). Such a comfortable balance would certainly attract attention as we are living in a world that values and protects data privacy.

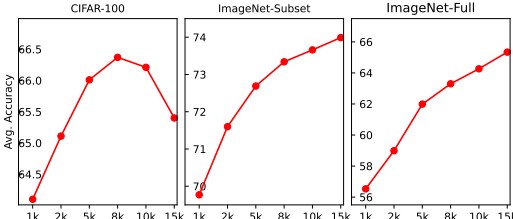

**Figure 3:** The impact of expansion size $d_{\text{(fe)}}$.

| FE process | w/ regularization | | | $\bar{\mathcal{A}}$ (%) |
|---|---|---|---|---|
| | $10^{-1}$ | $10^{-2}$ | $10^{-3}$ | |
| ✗ | ✓ | ✗ | ✗ | 52.99% |
| ✓ | ✓ | ✗ | ✗ | **66.30**% |
| ✓ | ✗ | ✓ | ✗ | 66.25% |
| ✓ | ✗ | ✗ | ✓ | 66.23% |
| ✓ | ✗ | ✗ | ✗ | 51.12% |

**Table 2:** Ablation study regarding expansion and regularization.

**Memory for Storage.** The ACIL stores $\boldsymbol{R}_k$ instead of exemplars. As an example, for fix-exemplar setting, the memory used by storing $\boldsymbol{R}_k$ ($8k$) on CIFAR-100/CUB200-2011/ImageNet is $8k \times 8k = 64$ million (M) tensor elements, while other methods consume 6.1M/301.1M/3010.6M respectively (e.g., on ImageNet $224 \times 224 \times 3 \times 20 \times 1000 \approx 3010.6\text{M}$). This shows that our method is memory-friendly to large-shaped image datasets (e.g., ImageNet).

### 4.4 Ablation Study

In the proposed ACIL, the FE process governed by $d_{\text{(fe)}}$ and the regularization controlled by $\gamma$ are essential. To show this, we adopt an ablation study by conducting a 5-phase CIL of ResNet-32 (with $d_{\text{(fe)}} = 8\text{k}$) on CIFAR-100 to observe the performance shift w.r.t. these modules. As reported in Table 2, without the FE process (see the first two rows in Table 2), the average incremental accuracy experiences a sharp drop (e.g., 66.30%→52.99%). The need for the FE process was enlightened by the fact that the analytic learning is naturally prone to under-fitting due to simple linear regression [29]. Widening the feature size helps to capture the missing discriminative information.

The regularization factor $\gamma$, on the other hand, plays an important role but behaves robustly during the CIL experiments. As shown rows 2-5 in Table 2, the ACIL performs robustly for a considerably wide range of $\gamma$ values (e.g., $10^{-3}$-$10^{-1}$). However, it would be unwise to remove the regularization as the ACIL could suffer from a strong accuracy reduction without its support (e.g., 66.30%→51.12%).

### 4.5   Potential Positive and Negative Societal Impacts

A main characteristic of ACIL is privacy-preserving. It allows researchers to avoid breaching user privacy while pushing forward their learning methods. From this angle, our method gives a positive societal impact by protecting privacy. We do not foresee obvious societal impacts. However, our ACIL does rely certainly on the CNN feature extractor in a similar way of transfer learning. In this case, there might be various follow-up research aiming to improve the extraction power by many trials and errors, leading to certain electricity pressure consumed by GPU operations.

## 5   Conclusion

In this paper, we have presented an analytic class-incremental learning (ACIL) which bears two valuable features (i.e., the absolute memorization and the data privacy protection) for addressing several existing limitations of class-incremental learning. The analytic learning has been incorporated as a key component to conduct incremental learning of new tasks in a recursive manner. Such a recursive learning style allows the ACIL to have absolute memorization. That is, the incremental learning of ACIL given present data would produce identical results to that of a joint learning which accesses both present and historical data, a property that has been theoretically validated. The recursive formulation has also the merit of not storing any samples from historical tasks, thus avoiding the breach of data privacy. Experiments have been conducted to validate our claims. Overall, our ACIL gives very competitive accuracy results. In particular, it outperforms the state-of-the-art methods for large-phase scenarios (e.g., incremental learning with 50 phases).

## 6   Acknowledgment

We thank the anonymous reviewers for their very constructive comments for improving this manuscript. This work was supported in part by the Science and Engineering Research Council, Agency of Science, Technology and Research, Singapore, through the National Robotics Program under Grant 1922500054.

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
