# Supplementary Material for ACIL: Analytic Class-Incremental Learning with Absolute Memorization and Privacy Protection

Huiping Zhuang[1], Zhenyu Weng[2*], Hongxin Wei[3], Renchunzi Xie[3], Kar-Ann Toh[4], Zhiping Lin[2]

[1]Shien-Ming Wu School of Intelligent Engineering, South China University of Technology, China
[2]School of Electrical and Electronic Engineering, Nanyang Technological University, Singapore
[3]School of Computer Science and Engineering, Nanyang Technological University, Singapore
[4]Department of Electrical and Electronic Engineering, Yonsei University, Korea
[1]hpzhuang@scut.edu.cn, [2]{zhenyu.weng, ezplin}@ntu.edu.sg
[3]{hongxin001, XIER0002}@e.ntu.edu.sg, [4]katoh@yonsei.ac.kr

## 1 Proof of Theorem

*Proof.* We first solves the recursive formulation for the RFAuM $\boldsymbol{R}_k$. According to the Woodbury matrix identity, for any invertible square matrices $\boldsymbol{A}$ and $\boldsymbol{C}$, we have

$$(\boldsymbol{A} + \boldsymbol{U}\boldsymbol{C}\boldsymbol{V})^{-1} = \boldsymbol{A}^{-1} - \boldsymbol{A}^{-1}\boldsymbol{U}(\boldsymbol{C}^{-1} + \boldsymbol{V}\boldsymbol{A}^{-1}\boldsymbol{U})\boldsymbol{V}\boldsymbol{A}^{-1}.$$

Let $\boldsymbol{A} = \boldsymbol{R}_{k-1}^{-1}$, $\boldsymbol{U} = \boldsymbol{X}_k^{\text{(fe)T}}$, $\boldsymbol{V} = \boldsymbol{X}_k^{\text{(fe)}}$, and $\boldsymbol{C} = \boldsymbol{I}$. Hence, from $\boldsymbol{R}_k = (\boldsymbol{R}_{k-1}^{-1} + \boldsymbol{X}_k^{\text{(fe)T}}\boldsymbol{X}_k^{\text{(fe)}})^{-1}$ and the Woodbury matrix identity, we have

$$\boldsymbol{R}_k = \boldsymbol{R}_{k-1} - \boldsymbol{R}_{k-1}\boldsymbol{X}_k^{\text{(fe)T}}(\boldsymbol{I} + \boldsymbol{X}_k^{\text{(fe)}}\boldsymbol{R}_{k-1}\boldsymbol{X}_k^{\text{(fe)T}})\boldsymbol{X}_k^{\text{(fe)}}\boldsymbol{R}_{k-1} \qquad \text{(a)}$$

which completes the proof for the recursive formulation of RFAuM. Let $\boldsymbol{Q}_{k-1} = [\boldsymbol{X}_0^{\text{(fe)T}}\boldsymbol{Y}_0 \ \dots \ \boldsymbol{X}_{k-1}^{\text{(fe)T}}\boldsymbol{Y}_{k-1}]$. According to (7), (8) and (a), we have

$$\hat{\boldsymbol{W}}_{\text{FCN}}^{(k)} = \boldsymbol{R}_k \begin{bmatrix} \boldsymbol{Q}_{k-1} & \boldsymbol{X}_k^{\text{(fe)T}}\boldsymbol{Y}_k^{\text{train}} \end{bmatrix}$$

$$= \begin{bmatrix} \boldsymbol{R}_k\boldsymbol{Q}_{k-1} & \boldsymbol{R}_k\boldsymbol{X}_k^{\text{(fe)T}}\boldsymbol{Y}_k^{\text{train}} \end{bmatrix} \qquad \text{(b)}$$

where

$$\boldsymbol{R}_k\boldsymbol{Q}_{k-1} = \boldsymbol{R}_{k-1}\boldsymbol{Q}_{k-1} - \boldsymbol{R}_{k-1}\boldsymbol{X}_k^{\text{(fe)T}}(\boldsymbol{I} + \boldsymbol{X}_k^{\text{(fe)}}\boldsymbol{R}_{k-1}\boldsymbol{X}_k^{\text{(fe)T}})^{-1}\boldsymbol{X}_k^{\text{(fe)}}\boldsymbol{R}_{k-1}\boldsymbol{Q}_{k-1}$$

$$= \hat{\boldsymbol{W}}_{\text{FCN}}^{(k-1)} - \boldsymbol{R}_{k-1}\boldsymbol{X}_k^{\text{(fe)T}}(\boldsymbol{I} + \boldsymbol{X}_k^{\text{(fe)}}\boldsymbol{R}_{k-1}\boldsymbol{X}_k^{\text{(fe)T}})^{-1}\boldsymbol{X}_k^{\text{(fe)}}\hat{\boldsymbol{W}}_{\text{FCN}}^{(k-1)}. \qquad \text{(c)}$$

Let $\boldsymbol{K}_k = (\boldsymbol{I} + \boldsymbol{X}_k^{\text{(fe)}}\boldsymbol{R}_{k-1}\boldsymbol{X}_k^{\text{(fe)T}})^{-1}$. Since,

$$\boldsymbol{I} = \boldsymbol{K}_k\boldsymbol{K}_k^{-1} = \boldsymbol{K}_k(\boldsymbol{I} + \boldsymbol{X}_k^{\text{(fe)}}\boldsymbol{R}_{k-1}\boldsymbol{X}_k^{\text{(fe)T}}),$$

we have $\boldsymbol{K}_k = \boldsymbol{I} - \boldsymbol{K}_k\boldsymbol{X}_k^{\text{(fe)}}\boldsymbol{R}_{k-1}\boldsymbol{X}_k^{\text{(fe)T}}$. Therefore,

$$\boldsymbol{R}_{k-1}\boldsymbol{X}_k^{\text{(fe)T}}(\boldsymbol{I} + \boldsymbol{X}_k^{\text{(fe)}}\boldsymbol{R}_{k-1}\boldsymbol{X}_k^{\text{(fe)T}})^{-1} = \boldsymbol{R}_{k-1}\boldsymbol{X}_k^{\text{(fe)T}}\boldsymbol{K}_k$$

$$= \boldsymbol{R}_{k-1}\boldsymbol{X}_k^{\text{(fe)T}}(\boldsymbol{I} - \boldsymbol{K}_k\boldsymbol{X}_k^{\text{(fe)}}\boldsymbol{R}_{k-1}\boldsymbol{X}_k^{\text{(fe)T}})$$

$$= (\boldsymbol{R}_{k-1} - \boldsymbol{R}_{k-1}\boldsymbol{K}_k\boldsymbol{X}_k^{\text{(fe)}}\boldsymbol{R}_{k-1})\boldsymbol{X}_k^{\text{(fe)T}} = \boldsymbol{R}_k\boldsymbol{X}_k^{\text{(fe)T}}$$

which allows (c) to be reduced to

$$\boldsymbol{R}_k\boldsymbol{Q}_{k-1} = \hat{\boldsymbol{W}}_{\text{FCN}}^{(k-1)} - \boldsymbol{R}_k\boldsymbol{X}_k^{\text{(fe)T}}\boldsymbol{X}_k^{\text{(fe)}}\hat{\boldsymbol{W}}_{\text{FCN}}^{(k-1)}. \qquad \text{(d)}$$

By substituting (d) into (b), we complete the proof. $\qquad \square$

---

[*]Corresponding author.

36th Conference on Neural Information Processing Systems (NeurIPS 2022).

# 2 Strict-Memory Setting

Here we also give the average incremental accuracy (see Table A) for the compared methods for strict-memory setting (i.e., only a fixed memory is allowed for the CIL). We adopt the memory budget used in the RMM paper [12]. In details, for each benchmark data, the memory budget is determined according to the phase number $K$. For instance [12], on CIFAR-10, the budget is 7k samples for $K = 5$ (7k samples = 10 classes per phase $\times$ 500 samples per class + 2k samples). The numbers reported in Table A are duplicated from [12] where the compared methods are implemented in the same setting.

The ACIL gives identical results either in growing-exemplar or fixed memory settings. This is because the ACIL does not belong to the branch of replay-based CIL.

Table A: Comparison of average incremental accuracy among compared methods for strict-memory setting.

| Metric | Method | Privacy | CIFAR-100 | | | | ImageNet-Subset | | | | ImageNet-Full | | | |
|---|---|---|---|---|---|---|---|---|---|---|---|---|---|---|
| | | | K=5 | 10 | 25 | 50 | K=5 | 10 | 25 | 50 | K=5 | 10 | 25 | 50 |
| $\bar{\mathcal{A}}(\%)$ | LwF (TPAMI 2018) | ✓ | 56.79 | 53.05 | 50.44 | - | 58.83 | 53.60 | 50.16 | - | 52.00 | 47.87 | 47.49 | - |
| | iCaRL (CVPR 2017) | × | 60.48 | 56.04 | 52.07 | - | 67.33 | 62.42 | 57.04 | - | 50.57 | 48.27 | 49.44 | - |
| | LUCIR (CVPR 2019) | × | 63.34 | 62.47 | 59.69 | - | 71.21 | 68.21 | 64.15 | - | 65.16 | 62.34 | 57.37 | - |
| | PODNet (ECCV 2020) | × | 64.60 | 63.13 | 61.96 | - | 76.45 | 74.66 | 70.15 | - | 66.80 | 64.89 | 60.28 | - |
| | LUCIR+Mnemonics (CVPR 2020) | × | 64.59 | 62.59 | 61.02 | - | 72.60 | 71.66 | 70.52 | - | 65.40 | 64.02 | 62.05 | - |
| | POD+AANets (CVPR 2021) | × | 66.61 | 64.61 | 62.63 | - | 77.36 | 75.83 | 72.18 | - | 67.97 | 65.03 | 62.03 | - |
| | POD+AANets+RMM (NeuriPS 2021) | × | **68.86** | **67.61** | **66.21** | - | **79.52** | **78.47** | **76.54** | - | **69.21** | **67.45** | 63.93 | - |
| | ACIL | ✓ | 66.30 | 66.07 | 65.95 | **66.01** | 74.81 | 74.76 | 74.59 | **74.13** | 65.34 | 64.84 | **64.63** | **64.35** |