# OpenReview forum: "ACIL: Analytic Class-Incremental Learning with Absolute Memorization and Privacy Protection"
_NeurIPS.cc/2022/Conference — NeurIPS 2022 Accept_

### Official Review · Reviewer_xNZm · 2022-07-05

**Rating:** 7
**Confidence:** 3
**Soundness:** 4 excellent
**Presentation:** 4 excellent
**Contribution:** 3 good

**Summary:**

This paper aims to tackle the problem of catastrophic forgetting without revisiting historical exemplars so that data privacy is preserved. To do this,  an analytic class-incremental learning (ACIL) algorithm is proposed with absolute memorization of past knowledge. Empirical results demonstrated competitive accuracy performance that does not degrade over increment of data classes during learning phases. . ACIL also outperformed the state-of-the-art methods for large-phase scenarios.

**Questions:**

1. Can you add descriptions of the benchmarking algorithms?
2. Why not evaluate the GAN-based methods on these large datasets?
3. Are the CNNs trained separately for each phase?

**Limitations:**

Not applicable.

**Strengths And Weaknesses:**

Strengths:
 - Section 3 clearly described the proposed algorithm. Even readers with no background in analytic learning and CIL can easily understand.
 - The idea of using analytic learning in CIL is novel and sound.
 - Has empirical evaluations demonstrating the power of ACIL.

Weaknesses:
 - Should explain the benchmarking algorithms in the empirical evaluations, e.g. what is LwF.
 - Should include more privacy-preserving baselines in the experiments. Currently, only LwF is used. How about the GAN-based methods mentioned in Section 2? Why not evaluate the GAN-based methods on these large datasets?
 - Should clarify how CNN was trained (separately for each phase or it is one CNN for every phase?)

---

> ### Author Response · Authors · 2022-08-01
> **Adding description of benchmarking algorithms, adding more privacy-preserving methods, and explaining how CNNs are trained**
>
> We thank the reviewer for the constructive comments! Our response is as follows.
>
> C1: Should explain the benchmarking algorithms in the empirical evaluations, e.g. what is LwF.
>
> Response: As suggested, we have included a brief explanation of the compared methods during the experiments. More detailed explanation can be found in the literature review (rows 67-98). The explanation is as follows.
>
> The LwF adopts distillation-based loss functions to prevent forgetting. The EWC uses Fisher information matrix. The SDC studies and compensates the semantic drift of features. The LwF, EWC, SDC and the proposed ACIL belong to the privacy-preserving methods. The other methods, i.e., BIC, iCaRL, Mnemonics, PODNet, AANets and RMM, are replay-based methods, requiring storage of past exemplars.
>
> The above discussion will be included in the revision (the manuscript has currently reached 9-page limit).
>
> C2: Should include more privacy-preserving baselines in the experiments. Currently, only LwF is used. How about the GAN-based methods mentioned in Section 2? Why not evaluate the GAN-based methods on these large datasets?
>
> Response: As suggested, we have included two more privacy-preserving baselines, i.e., EWC [R3-1] and SDC [R3-2]. The results have been included in Table 1 and Figure 2 (please see the revised manuscript, sorry cannot paste big table and figure here).
>
> Regarding the GAN-based CIL, training GANs on large datasets in this paper (e.g., ImageNet) is very time-consuming and unstable, and (to the authors’ knowledge) it has not been implemented by any other CIL papers yet likely owing to the same concern. More importantly, GAN-based methods rely heavily on GAN training, yet the training requires very delicate tuning of hyperparameters. It might not be fair to conduct such a comparison (without putting huge amount of effort). However, if the reviewer believes the GAN-based methods are necessary to improve this manuscript, we shall try our best to conduct a comparison to make it in the reviewer-author discussion.
>
>
> [R3-1] Kirkpatrick, J., Pascanu, R., Rabinowitz, N., Veness, J., Desjardins, G., Rusu, A. A., ... & Hadsell, R. (2017). Overcoming catastrophic forgetting in neural networks. Proceedings of the national academy of sciences, 114(13), 3521-3526.
>
> [R3-2] Yu, Lu, Bartlomiej Twardowski, Xialei Liu, Luis Herranz, Kai Wang, Yongmei Cheng, Shangling Jui, and Joost van de Weijer. "Semantic drift compensation for class-incremental learning." In Proceedings of the IEEE/CVF Conference on Computer Vision and Pattern Recognition, pp. 6982-6991. 2020.
>
>
> C3: Should clarify how CNN was trained (separately for each phase or it is one CNN for every phase?)
>
> Response: We thank the reviewer for the constructive comment. The CNN is only trained during the BP base training phase. After that, the parameters of the CNN backbone are fixed during the incremental phases (i.e, phase #1 to #K) including the ARaBT (Analytic Re-alignment Base Training).
>
> The above explanation will be included in the revision (the manuscript has currently reached 9-page limit).

---

### Official Review · Reviewer_UFfv · 2022-07-13

**Rating:** 7
**Confidence:** 4
**Soundness:** 3 good
**Presentation:** 3 good
**Contribution:** 3 good

**Summary:**

In this paper, the authors propose a method to learn a perfect FC classifier for CIL without using exemplars. Instead, they need to store autocorrelation matrixes in the memory. They provide extensive experiments to show the effectiveness of the proposed method.

**Questions:**

See the "weaknesses" part.

**Limitations:**

It would be better to add a section to discuss the potential negative societal impact.

**Strengths And Weaknesses:**

### Strengths

- This paper is well-written and easy to follow.

- The proposed method achieves impressive results without using the exemplars.

- Extensive experiment results on multiple datasets are provided.

### Weaknesses

I have reviewed this paper as an ICML submission. During the review, I indicate the following weaknesses:

- In the abstract and introduction, the authors claimed they fully address the catastrophic forgetting problem. However, the results (e.g., in Table 1) of ACIL are still much lower than the joint training results. I think it is because CNN is fixed. Therefore, the authors should adjust their presentation and remove the claim of “fully addressing the catastrophic forgetting”.

- The proposed method needs to store autocorrelation matrixes in the memory. Thus, the author should also compare the memory usage between existing methods and the proposed method. The authors need to answer whether the proposed method requires more memory compared to saving exemplars.

- The proposed method uses a fixed CNN network after the first phase. So it is not possible to update the low-level image features with the new data in the following phases. If we observe some new classes that are very different from the first phase data, this method might not perform well. Maybe the authors should include some experiments to evaluate the above situation.

- There are some small typos in the paper. For example, the authors write “
Merits of ACL come with costs” in Line 014. However, the abbreviation “ACL” never appears before. I think it might be “CIL”. In Figure 1, the word “clasasifier” is incorrect.

- The authors should move the proof of Theorem 3.1 to the supplementary.

The above weaknesses have been addressed in the current version. Thus, I have no more questions and recommend acceptance.

---

> ### Author Response · Authors · 2022-08-01
> **Thanks for the constructive comments that have improved our mansucript!**
>
> C1: I have reviewed this paper as an ICML submission. During the review, I indicate the following weaknesses. The above weaknesses have been addressed in the current version. Thus, I have no more questions and recommend acceptance.
>
> Response: We thank the reviewer for the constructive comments! They have been very useful for improving our manuscript!
>
> C2: It would be better to add a section to discuss the potential negative societal impact.
>
> Response: As suggested, we have discussed the potential positive and negative societal impacts as follows.
>
> A main characteristic of ACIL is privacy-preserving. It allows researchers to avoid breaching user privacy while pushing forward their learning methods. From this angle, our method gives a positive societal impact by protecting privacy. We do not foresee obvious societal impacts. However, our ACIL does rely certainly on the CNN feature extractor in a similar way of transfer learning. In this case, there might be various follow-up research aiming to improve the extraction power by many trials and errors, leading to certain electricity pressure consumed by GPU operations.
>
> The discussion will be included in the revision (the manuscript has currently reached 9-page limit).

---

> > ### Comment · Reviewer_UFfv · 2022-08-01
> > **Thanks for the feedback**
> >
> > Thanks for the feedback from the authors. I decide to keep my initial rating "accept". Please consider adding the discussion on potential negative societal impact in the final version.

---

> > > ### Author Response · Authors · 2022-08-03
> > > **Thanks**
> > >
> > > Thank you! We will include the potential negative societal impact in the final version as the reviewer suggested!

---

### Official Review · Reviewer_B6wm · 2022-07-17

**Rating:** 8
**Confidence:** 3
**Soundness:** 3 good
**Presentation:** 4 excellent
**Contribution:** 3 good

**Summary:**

The paper suggests a new CIL (Class-incremental learning) algorithm. The proposed algorithm outperforms existing CIL algorithms in the sense that it is both reasonably accurate, and does not expose past examples to privacy breaches. Previous algorithms could have only achieved one of these desired properties.

**Questions:**

Is the conjecture that the mild degradation mentioned in row 263 could be explained by quantization errors supported by the reference cited there? What is the reasonable damage that such errors usually cause?

**Limitations:**

Yes.

**Strengths And Weaknesses:**

Strengths:
1. The key contribution of an algorithm which is both accurate and private seems novel and significant. It is also well explained and motivated.
2. A thorough literature review is conducted, assisting the reader to understand the paper's contribution.
3. The limitations of the algorithm (especially for small K values) are well explained.

Weaknesses:
1. It could have been interesting to analyze why for small K values ACIL does not outperform state-of-the-art, and perhaps even to conjecture whether there exists a private algorithm that outperforms the state-of-the-art for any value of K.
2. For the inexperienced reader, it is not obvious that the mild degradation mentioned in row 263 could really be explained by quantization errors, as he/she may not be familiar with the plausible rate of such kind of errors. Also, it might have been interesting to discuss what could happen when K is extremely large (in case that it is indeed an interesting scenario, and if it is not, it might be worth explaining why). As far as the (inexperienced) reader knows, the degradation might be catastrophically high in this case.

---

> ### Author Response · Authors · 2022-08-01
> **Explaining why ACIL performs less idealy in small-phase tasks, and the conjecture of quantization errors**
>
> We thank the reviewer for the constructive comments! Our response is as follows.
>
> C1: It could have been interesting to analyze why for small K values ACIL does not outperform state-of-the-art, and perhaps even to conjecture whether there exists a private algorithm that outperforms the state-of-the-art for any value of K.
>
> Response: For small K values, i.e., small-phase tasks, it is natural that conventional CIL methods (e.g., LUCIR) experience much less forgetting (hence better performance) than that of large-phase ones. Our ACIL does not forget historical knowledge (regardless of small- or large-phase tasks). This is at the cost of freezing the trained CNN backbone during the base phase, leading to certain performance decline. Such a decline can be treated as a K-invariant “overhead”. For small K scenario, this “overhead” accuracy loss is larger than that from those conventional methods. However, as K increases, the constant “overhead” loss becomes a reasonable cost as conventional methods would suffer much more greatly. The fact that our ACIL is also equipped with privacy-preserving property further indicates the strength of the proposed method.
>
> Regarding the conjecture for a private algorithm that outperforms the SOTA methods, we have two opinions. One is that, it is our goal (future work) to construct privacy-preserving algorithms to beat the current SOTA performance (i.e., RMM method) for any K values, and we do believe its possibility. The other opinion is that, from the information point of view, privacy-preserving methods (e.g., ACIL) forgo useful data compared with privacy-invading methods. The performance of the privacy-preserving methods should naturally be slightly worse (at least for small K values). From another angle, our ACIL can catch up with the existing SOTA, suggesting that the exemplar-based methods have not hit their limits yet.
>
>
>
> C2: For the inexperienced reader, it is not obvious that the mild degradation mentioned in row 263 could really be explained by quantization errors, as he/she may not be familiar with the plausible rate of such kind of errors. Is the conjecture that the mild degradation mentioned in row 263 could be explained by quantization errors supported by the reference cited there? What is the reasonable damage that such errors usually cause?
>
> Response: Our method is, in essence, recursive least squares based. As the recursion number (i.e., K) increases, more computation steps are required. Each computational step leads to a round of quantization, resulting in a slight shift of classification boundaries. This has been pointed out in reference [R1-1]. For training on relatively large datasets (e.g., CIFAR or larger, see TABLE VI in [R1-1]), even for a shallow fully-connected network, there is obvious classification difference (0.4%) between the classifiers trained with small and large number of recursive steps. Datasets used in this paper are significantly larger (e.g., ImageNet) and networks have more parameters. Hence, our conjecture of attributing the mild accuracy degradation to quantization is reasonable. Also, as indicated in [R1-1], though quantization errors exist, they only affect the classification results of examples residing near the classification boundaries, hence only giving very limited impact on the classification accuracy.
>
> C3: Also, it might have been interesting to discuss what could happen when K is extremely large (in case that it is indeed an interesting scenario, and if it is not, it might be worth explaining why). As far as the (inexperienced) reader knows, the degradation might be catastrophically high in this case.
>
> Response: Regarding extremely large K, for our ACIL, theoretically the performance should stay the same. To support our claim, we have managed to conduct an extra 250-phase (with an expansion size of 5k) experiment on ImageNet (see below).
>
> ACIL K=5.     Avg. acc = 61.99%
>
> ACIL K=250  Avg. acc = 61.53%
>
> The results have again empirically validated our claim of “absolute memorization” as the average accuracy for 250-phase CIL is almost the same as that for 5-phase CIL.
>
> [R1-1] Huiping Zhuang, Zhiping Lin, and Kar-Ann Toh. "Blockwise Recursive Moore–Penrose Inverse for Network Learning." IEEE Transactions on Systems, Man, and Cybernetics: Systems 52, no. 5 (2022): 3237-3250.

---

> > ### Comment · Reviewer_B6wm · 2022-08-08
> > **Response to authors**
> >
> > Thank you very much for responding to my review. Your comment really clarified the primary issues I was not sure about.
> > It is really nice that you managed to run an experiment with K=250 to support your claim for large K values! Thanks for that.

---

> > > ### Author Response · Authors · 2022-08-08
> > > **Thanks!**
> > >
> > > Thank you very much for the response! Your comments are very helpful!

---

### Meta-Review · Area_Chair_fJmw · 2022-08-25

**Recommendation:** Accept
**Confidence:** Certain

**Metareview:**

The reviewers agree that this is a solid contribution. Please do revise the paper according to the reviewers comments and the discussion.

**Award:**

No

---

### Decision · Program_Chairs · 2022-09-14

Accept